# On the Torsional Behavior of the Longitudinal Bridge Girders Used in the LT-Bridge Construction Method

**Michael Rath \*, Franz Untermarzoner and Johann Kollegger**

Institute of Structural Engineering, TU Wien, 1040 Vienna, Austria; franz.untermarzoner@tuwien.ac.at (F.U.); johann.kollegger@tuwien.ac.at (J.K.)
\* Correspondence: michael.rath@tuwien.ac.at

**Abstract:** A new bridge construction method, combining semi-precast elements and in situ concrete, has been developed at TU Wien, with the aim of decreasing erection time. In the course of construction using this new method, structural conditions arise that render a more detailed investigation necessary. By connecting a precast, thin-walled box girder to a bridge segment located on a pier by means of post-tensioning, a joint is created. By casting in situ concrete on the bottom and top slabs, the joint can be bridged with longitudinal reinforcement; however, the unreinforced vertical joints in the webs remain. This detail is a specific characteristic of the LT-bridge construction method and needs to be further investigated and assessed, as the question arises as to how this circumstance affects the torsional bearing behavior of the bridge superstructure. Torsion tests described in the literature consider ordinary box girders with longitudinal reinforcement or post-tensioned segmental bridges without longitudinal reinforcement at the joints. Therefore, the new reinforcement layout at the joints had to be investigated experimentally. Two large-scale thin-walled box girders—one without joints in the webs and the other with unreinforced joints in the webs—were tested, allowing for a direct comparison of conventionally manufactured bridges and those erected with the new bridge construction method. Furthermore, we investigated whether the results of common calculation methods corresponded to the experimental findings.

**Keywords:** concrete bridge construction; hollow box girder; new LT-bridge construction method; thin-walled precast elements; torsion; torsional testing; shear joints





## 1. Introduction

The use of thin-walled prefabricated concrete elements in bridge engineering has played a major role in the relevant research activities of the Institute of Structural Engineering of TU Wien in recent years [1,2]. The implementation of such lightweight structural elements allows for the development of new approaches for the construction of bridges, as was impressively demonstrated when the balanced lowering method [3–5] was used to build two bridges over the Lafnitz and Lahnbach Rivers in Austria. Another application concerns the semi-precast segmental bridge construction method [6–8], in which thin-walled precast concrete elements are assembled to form a box girder with the aid of steel components.

The latest development to emerge from this research area is the new LT-bridge construction method, which combines hollow box girders and special deck slab elements (both components made of thin-walled precast elements), focusing yet again on erecting bridges in a very short construction time [9,10]. The term "LT-bridge construction method" originates from the span directions of the individual elements, with the hollow box girders spanning longitudinally (L) and the deck slab elements spanning transversally (T). Currently, the method has only been developed for single-cell box girders. This paper focuses on the torsional bearing behavior of the resulting bridge superstructure, as it presents peculiarities that require a detailed scientific investigation. The uniqueness of the finished

bridge superstructure stems from the fact that it has continuous deck and bottom slabs, with regard to the longitudinal direction of the bridge, while simultaneously working with vertical shear joints without longitudinal reinforcement in the webs. This circumstance results directly from the bridge construction process, which is depicted in Figure 1a. A hollow box girder with a wall thickness of around 100–200 mm is lifted in between two already-placed pier segments. In this construction phase, horizontal and vertical joints are still existent between the girder and pier segments. Within the next construction step, special deck slab elements are placed on top of the previously mounted girder. With the use of in situ concrete layers, the deck and bottom slabs attain their final thicknesses. This subsequently cast concrete allows for the placement of continuous longitudinal reinforcement in the deck and bottom slabs, whereby rebar couplers are used in the latter (Figure 1b). Furthermore, post-tensioning is used to connect the girder to the pier segment. The result is a bridge superstructure with unreinforced joints in the webs at the connection to the pier segment and continuous longitudinally reinforced bottom and deck slabs. The LT-bridge construction method is therefore the motivation for the investigation of box girders with this unique reinforcement arrangement at the joints.

The purpose of this paper was to analyze how the unreinforced joints in the webs affect the torsional load-bearing behavior. Compared to a superstructure of a bridge erected with the LT-bridge construction method, simplifications—which are discussed in more detail in Section 2—were made. As the investigation was seen as fundamental, post-tensioning was omitted. This resulted in the shear joints not being under longitudinal compressive stresses, possibly reducing the transmission of shear stresses. In further test series, the influence of post-tensioning will be considered in detail, including long-term effects, e.g., as described by Huang et al. [11]. The main objective of the research presented in this paper was to gain an understanding of how reinforced concrete box girders resist torsion when unreinforced joints are present in the webs. Of particular interest is the comparison to the behavior of regular girders with continuous longitudinal reinforcement throughout the entire cross-section. Furthermore, it was important to clarify which calculation methods can accurately describe the occurring behavior and, thus, whether it is covered by normative standards. In order to obtain answers to these questions, large-scale experimental torsion tests replicating the conditions described above were carried out in the laboratory of the Institute for Structural Engineering of TU Wien.

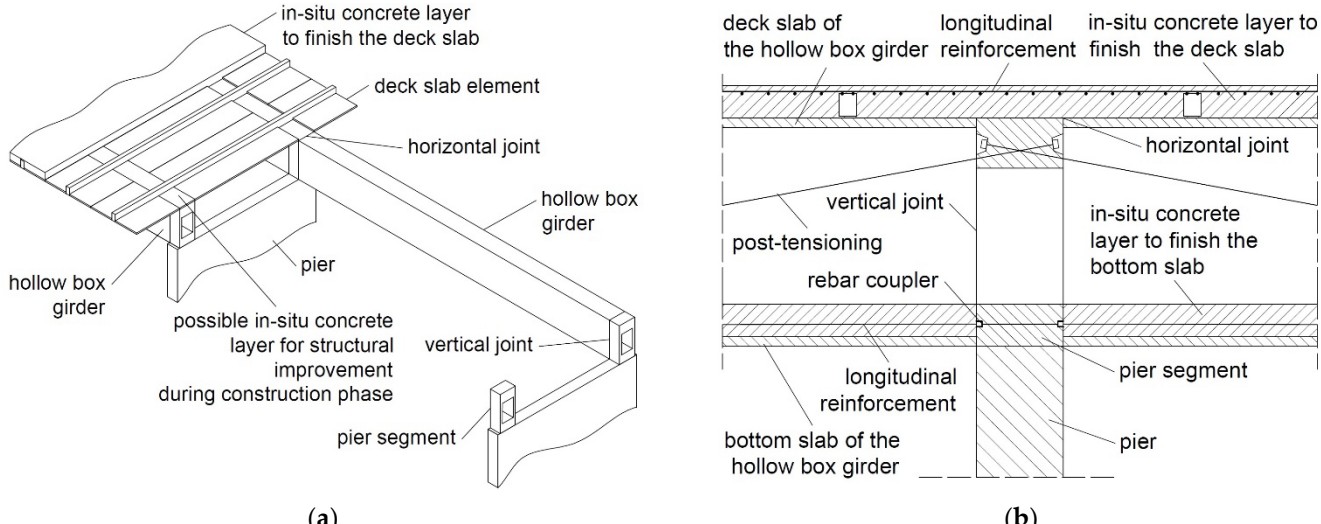

**(a)**          **(b)**

**Figure 1.** Concept of the new LT-bridge construction method (drawings based on [12]): (**a**) construction process, showing the resulting horizontal and vertical joints; (**b**) longitudinal section over the pier, highlighting the continuous reinforcement in the deck and bottom slabs due to the in situ layers.

It is also conceivable that the hollow box girder itself consists of several segments that are connected to one another by post-tensioning. This would result in further joints without continuous reinforcement in the webs. However, the connection to the pier segment is subjected to vertical compressive stresses resulting from the vertical part of the anchorage forces of the post-tensioning tendons (Figure 1b), as well as from self-weight and torsion at the support (pier). This circumstance was taken into account in the experimental test program, with an investigation of joints without vertical compression of the webs planned in further related test series.

Typical torsional reinforcement is composed of longitudinal reinforcement and stirrups with the individual load-bearing distribution described by Mitchell and Collins [13,14]. The load transfer is characterized by a strut-and-tie model after crack formation (Eurocode 2 [15]). The distribution of the longitudinal reinforcement along the perimeter of the cross-section can be arranged in different ways. It is therefore possible to either distribute the reinforcement evenly around the perimeter or to concentrate it in the corners of the box girder. This is confirmed by the test series described in detail and performed by Lampert et al. [16–19], where these two reinforcement arrangements led to different crack and deformation behaviors during loading but showed no difference in the ultimate torsional moment [20]. Due to peculiarities of the LT-bridge construction method, a concentrated, continuous torsional reinforcement cannot be located exactly in the corners of the cross-section. The continuous longitudinal reinforcement in the bottom slab—crossing the joint to the pier segment—can only be embedded in the in situ concrete layer. As a result, the concentrated corner reinforcement has a lateral offset. This circumstance, along with the unreinforced vertical joints in the webs, leads to the essential differences between the experimental torsion tests described here and those presented by Lampert [20]. Furthermore, a large-scale test with a thin-walled box girder cross-section was carried out during the development of the balanced lift method, but joints in the cross-section were not considered [21]. A comprehensive review of torsion tests found in the literature is presented by Humer [22]. In addition, various torsion tests have been carried out on hollow box girders in recent years, e.g., as presented by Lopes and Bernardo [23], where high-strength concrete was used and the ductility behavior was studied in more detail. Further investigations by Jeng et al. [24] and Specker [25] verified the so-called "Softened Membrane Model Theory" and assessed the influence of joints on the torsional capacity of externally prestressed box girders, respectively. The latter analyzed girders both with and without unreinforced joints. The unique arrangement of unreinforced joints occurring only in the webs—as is the case within the LT-bridge construction method—therefore requires a new experimental investigation, which is presented from Section 2 onwards.

In general, concrete struts resulting from torsional loading occur at an angle of 45° to the longitudinal axis of the girder if the design is set on yielding of both longitudinal and stirrup reinforcement at the same torsional moment. If this is not the case, the inclination of the compression struts $\theta$ deviates from the 45° angle, as documented by Lampert [20]. The forces are redistributed to the unyielding components (i.e., either the longitudinal reinforcement or the stirrups), leading to a subsequent change in the inclination of the concrete compression struts. This phenomenon is highly relevant for the recalculation of the experimental investigations described in Section 3.

Naturally, the applied shear force and the bending moment also have an influence on the torsional load-bearing capacity. As such, the additional shear stresses must be considered in the design of the stirrups, while the longitudinal forces resulting from the bending moment are added to those from the torsional moment. Additional reinforcement is therefore required on the side subjected to tension, while the forces resulting from torsion counteract on the compression side.

## 2. Experimental Investigations on the Torsional Behavior of Hollow Box Girders with Joints in the Webs

### 2.1. Basic Concept of the Experimental Program

The experimental investigations considered two large-scale test specimens, resulting in four possible torsion tests with the selected test setup. Both test specimens had a total length of 9.20 m, cross-sectional dimensions of 1.30 m by 1.00 m (height to width), and a wall thickness of 100 mm. The basic concept of the experimental investigations was to conduct a torsion test with a static system (Figure 2a,b) of a cantilever with a length of 4.60 m (corresponding to half the length of the test specimens), allowing two tests per test specimen. By applying the torsional load at the end of the test specimen (cantilever end) and creating a restraint in the middle to absorb the applied load, it was possible to leave the other half of the test specimen undamaged. Through modification of the test setup or rotation of the test specimen, a second torsion test was then performed on the undamaged half.

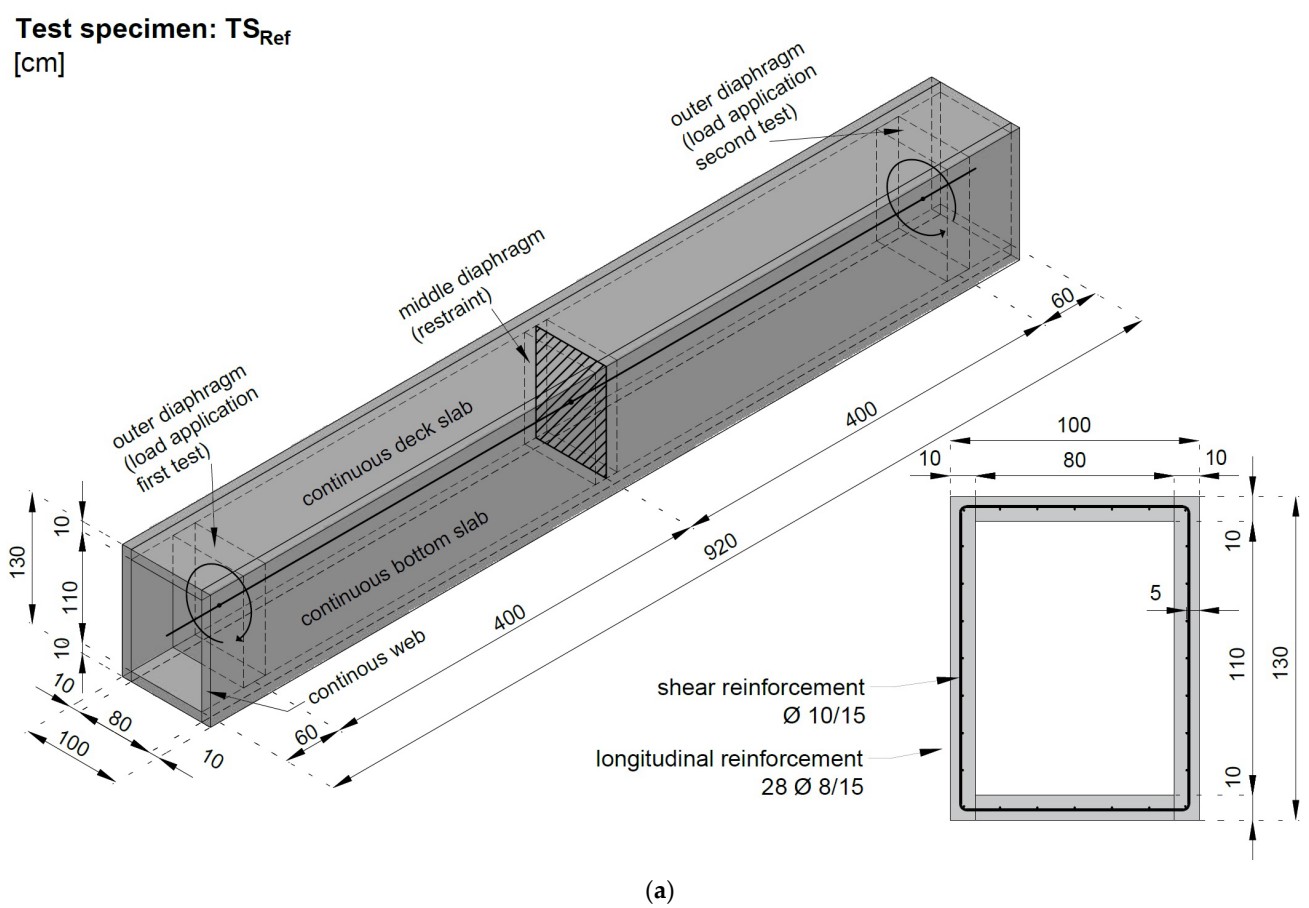

(**a**)

**Figure 2.** *Cont.*

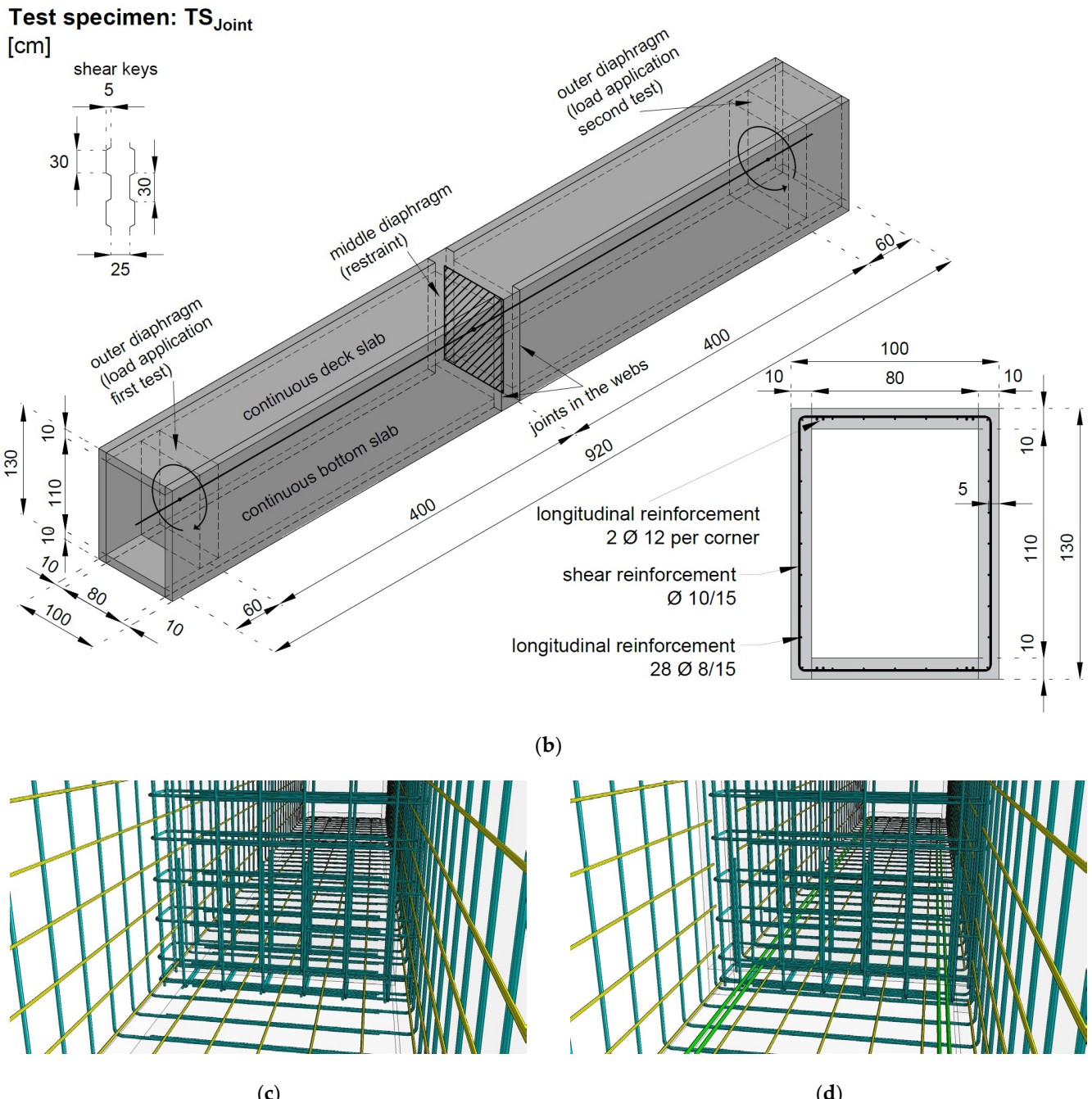

**Figure 2.** Overview, dimensions, and reinforcement layout of the two test specimens: (**a**) static system, dimensions, and cross-section of $TS_{Ref}$; (**b**) static system, dimensions, and cross-section of $TS_{Joint}$; (**c**) reinforcement details for $TS_{Ref}$, showing the bottom slab, webs, and middle diaphragm. Yellow: Ø8 continuous longitudinal reinforcement in bottom slab and webs, cyan: Ø10 stirrups; (**d**) reinforcement details for $TS_{Joint}$, showing the bottom slab, webs, and middle diaphragm. Yellow: Ø8 continuous longitudinal reinforcement in the bottom slab and longitudinal reinforcement abutting the joint in the webs; green: Ø12 continuous longitudinal reinforcement in the bottom slab; cyan: Ø10 stirrups.

One of the two test specimens was a hollow box girder with continuous longitudinal torsion reinforcement without joints in the webs (named $TS_{Ref}$; shown in Figure 2a,c), while the other represented the unique connection between the hollow box girder and the pier segment of the novel bridge construction method (named $TS_{Joint}$; shown in Figure 2b,d).

The test bodies were fitted with a total of three diaphragms with a thickness of 0.25 m each; while the outer two were necessary for the safe introduction of the torsional loads, the middle diaphragm was required for the restraint, which was achieved by applying a large vertical force to counteract the initiated loads. In the case of $TS_{Joint}$, the middle diaphragm—with a height of 1.10 m and width of 1.00 m—additionally represented the pier segment of the LT-bridge construction method, with the unreinforced joints situated between the webs and the middle diaphragm itself. The bottom and top slabs of $TS_{Joint}$ were cast continuously over the whole length of 9.20 m. The webs, bottom slab, and deck slab of $TS_{Ref}$, on the other hand, were cast continuously over the whole length of 9.20 m, which was made possible by reducing the width of the middle diaphragm to 0.80 m.

According to Lampert et al. [18], post-tensioning tendons can be considered as reinforcement in terms of torsional resistance if they are bonded; however, careful considerations have to be made regarding the external post-tensioning used in the new construction method. Considering this fact, it was decided to omit the post-tensioning for the time being, which also led to a simpler test procedure and a more conclusive evaluation of the results. The effects of external post-tensioning will be evaluated in a further test series. Due to the anchoring of the post-tensioning tendons at the pier segment, vertical compressive stresses occur, which also locally influence the webs and, thus, the joint. This vertical pressure was automatically generated in the test program when the vertical loads on the middle diaphragm (representing the pier segment) were applied for the creation of the restraint. The force was selected in such a way that the resulting compressive stress in the webs and joint approximately corresponded to a realistic bridge design. The exact test setup is discussed in more detail in Section 2.2.

### 2.1.1. Test Specimen without Joints in the Webs ($TS_{Ref}$)

The purpose of this test specimen was to serve as a reference girder, in the sense that it was a classically reinforced box girder without joints and with continuous longitudinal reinforcement—both in the bottom and top slabs and in the two webs—facilitating comparison with the bridge superstructure of the new LT-bridge construction method.

The longitudinal reinforcement had a diameter of 8 mm, with a spacing of 150 mm between the bars. One bar was always placed exactly in each corner of the box girder. The longitudinal reinforcement had a concrete cover of 46 mm; therefore, it was located along the centerline of the webs, deck slab, and bottom slab. The stirrups surrounding the longitudinal reinforcement had a diameter of 10 mm, with a spacing of 150 mm. Due to the manufacturing process of the test specimens, the stirrups consisted of two U-shaped reinforcement elements, which overlapped across the entire cross-sectional width of the bottom and top slabs.

The construction process consisted of several steps, as shown in Figure 3. First, the two webs with a length of 9.20 m were cast in a horizontal position. Figure 3a shows the formwork before concreting, with the longitudinal reinforcement and the U-shaped stirrup reinforcements visible. In the next step, the two webs were set up vertically, and the reinforcement for the bottom slab and the 0.25 m thick middle diaphragm was installed (Figure 3b). The bottom slab between the webs was then cast continuously over the length of 9.20 m. The top slab between the webs and the middle diaphragm was cast in one pour (Figure 3c). The finished test specimen is shown in Figure 3d. The production of the two 0.25 m thick outer diaphragms was carried out subsequently, as otherwise the formwork of the deck slab could not have been removed. The position of these diaphragms was chosen in such a manner that their centerline was 4.00 m from the center of the girder, based on the equipment of the laboratory where the experimental investigations were carried out. The formwork for the outer diaphragms was produced and wedged in the finished box girder. Fast-setting grout with a compressive strength of 30 N/mm$^2$ was used to cast the diaphragms, using filler holes in the top slab. The reinforcement consisted of formwork anchors that were guided through prepared holes around the cross-section and anchored from the outside. The inner formwork wall remained in the finished test specimen.

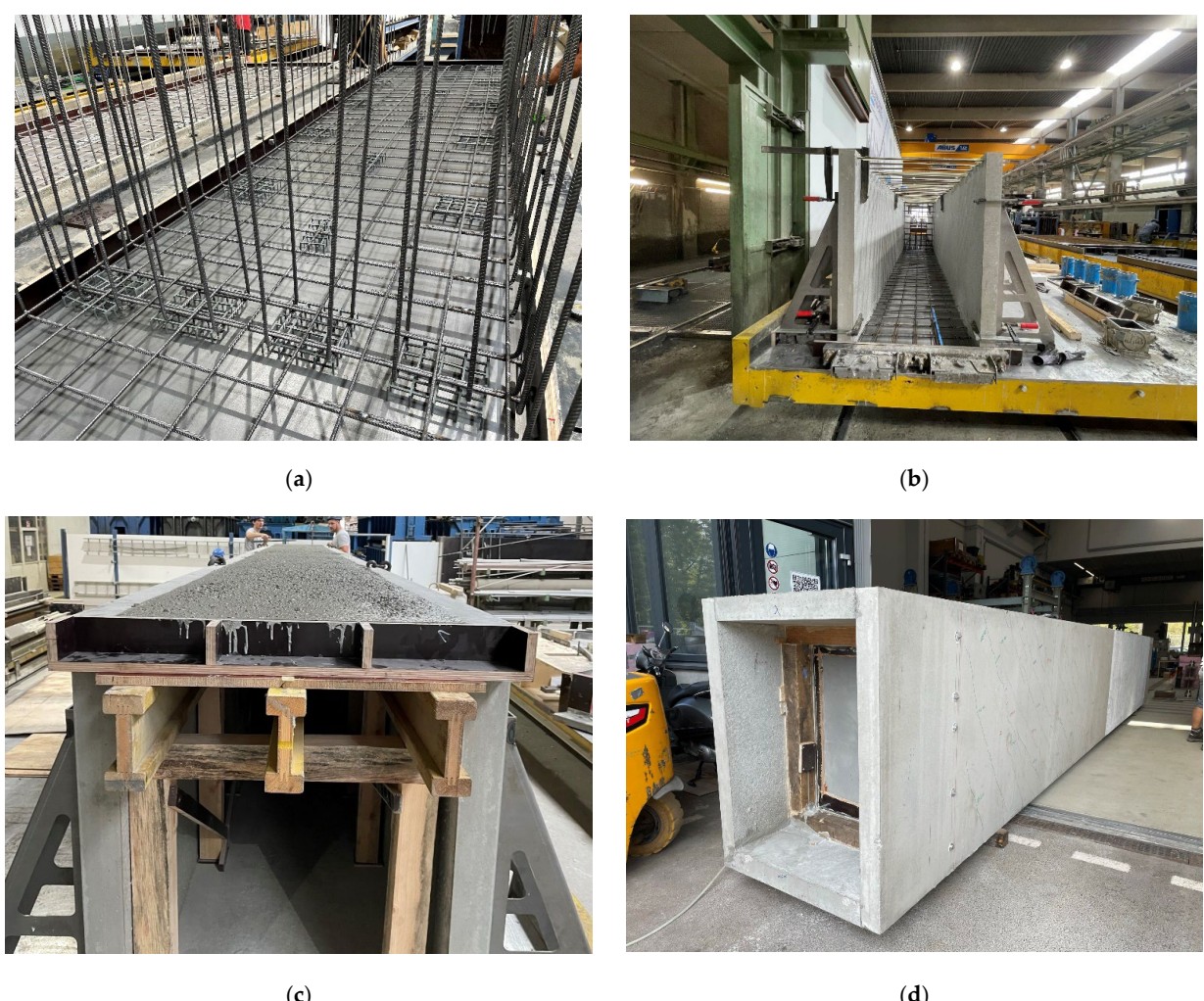

(**a**)

(**b**)

(**c**)

(**d**)

**Figure 3.** Construction process of the test specimen serving as the reference girder ($TS_{Ref}$): (**a**) reinforcement of the 9.20 m long webs and parts of the middle diaphragm before casting; (**b**) reinforcement of the 9.20 m long bottom slab between the webs before casting; (**c**) casting of the deck slab and the diaphragm in the middle of the girder; (**d**) completed test specimen.

2.1.2. Test Specimen with Unreinforced Joints in the Webs ($TS_{Joint}$)

In contrast to the reference girder described above, this test specimen had unreinforced joints in the webs, representing the bridge superstructure of the presented bridge construction method. Compared to the actual bridge superstructure, the following simplifications—in addition to the ones described in the Introduction—were made for the test program: Instead of producing an in situ concrete layer on the bottom and top slabs and placing the continuous longitudinal reinforcement there, it was decided to produce the bottom and top slabs as a whole in one pour. Compared to the real bridge superstructure, the reinforcement in the vertical position could thus be laid somewhat closer to the outer edge of the concrete, while the horizontal distance to the corner remained consistent.

The test specimen was designed with the same amount of continuous longitudinal reinforcement as the reference beam. As the longitudinal reinforcement of the webs should not be effective due to the joints in the webs as torsional reinforcement, an equivalent amount of additional longitudinal reinforcement was placed in the bottom and top slabs. This meant that, in addition to the Ø8/150 of the reference girder, two Ø12 bars were placed in each corner of the deck and bottom slabs. The position of this additional reinforcement was chosen to be as close as possible to the edge of the corners. In the actual bridge superstructure with in situ concrete layers, this would be the case for the bottom slab (apart

from the abovementioned vertical distance). The total area of continuous longitudinal reinforcement in this girder, with 10 Ø8 mm and 8 Ø12 mm (total 14.074 cm$^2$), was equal to that of the reference specimen with 28 Ø8 mm.

The production process of the specimen with joints (Figure 4) was essentially the same as for the reference beam, with the only difference being that the webs were not produced in one step. Instead, each web was divided into two partial plates, which were concreted separately (Figure 4a depicts the unreinforced joints between the webs). At the ends of the slabs, where the joints were to be created, shear keys—based on [7]—with a depth of 5 mm and a height of 30 mm were placed over the entire height of the web (Figure 2b). The sections with a length of 4.475 m were then moved to a vertical position, with a 0.25 m gap left between them for the middle diaphragm, representing the pier segment. As previously described, the unreinforced vertical shear joints in the webs were located at the transition between the webs and the middle diaphragm, as would be the case with a pier segment and the hollow box girder in the novel bridge construction method. In contrast to the joints studied by Fasching et al. [7], no special mortar was used for filling the joints, as the concrete of the diaphragm itself served this purpose. Before the diaphragm was cast, the entire bottom slab with a length of 9.20 m was concreted between the webs (Figure 4b shows the reinforcement of the bottom slab and middle diaphragm, as well as the joint). Afterwards, the middle diaphragm was cast together with the deck slab in one step (Figure 4c shows the reinforcement before casting). As was the case for the reference girder, concreting of the outer diaphragms was carried out subsequently, at a distance of 4.00 m from the center of the girder, using a fast-setting grout. Figure 4d shows the final test specimen.

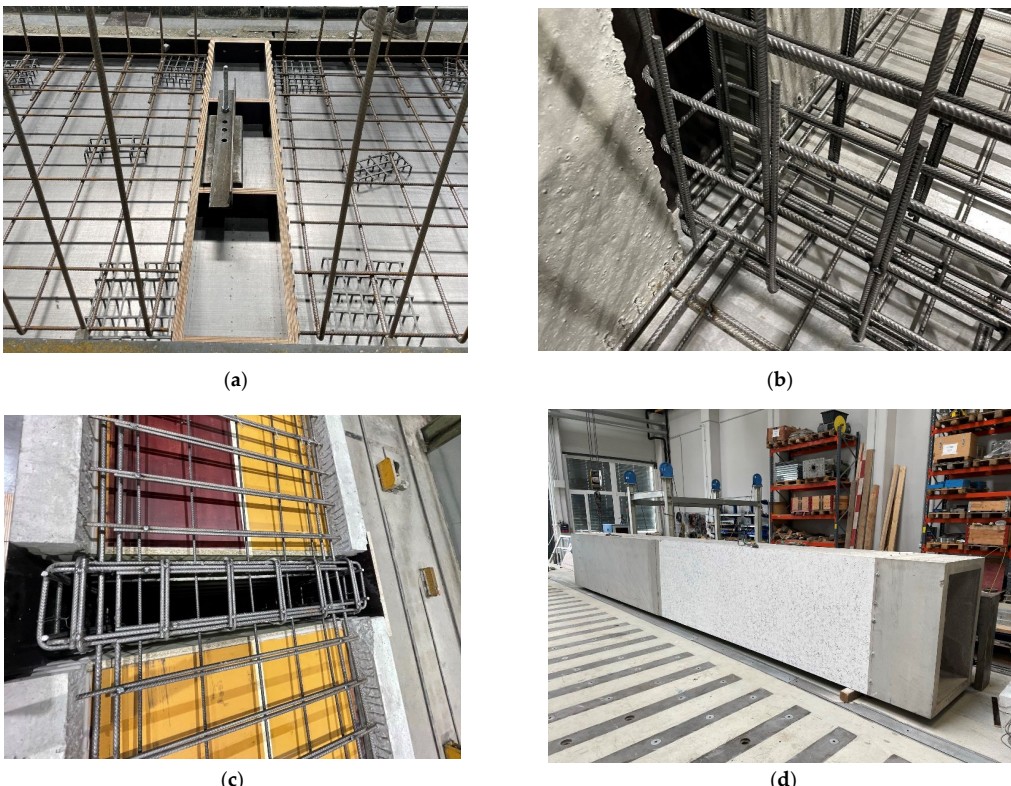

|         |         |
| :-----: | :-----: |
| (**a**) | (**b**) |
| (**c**) | (**d**) |

**Figure 4.** Construction process of the test specimen serving as a representation of the superstructure of the novel bridge construction method (TS$_{Joint}$): (**a**) reinforcement of the four parts of webs with a length of 4.475 m, showing the shear joints before casting; (**b**) reinforcement of the 9.20 m bottom slab and middle diaphragm, highlighting the continuous reinforcement in the bottom slab and shear joints in the webs before casting; (**c**) reinforcement of the deck slab and the diaphragm in the middle of the girder before casting; (**d**) finished test specimen.

### 2.1.3. Materials

As the two test specimens had to be produced over several days, considering the manufacturing steps, the individual components had slightly different material properties. In general, the aim was to achieve an average cylinder compressive strength $f_{cm,cyl}$ of at least 48 N/mm$^2$ (C40/50). Concrete with a maximum grain size of 16 mm was compacted with the aid of a vibrating table and, if necessary, hand vibrators. A list of the various properties—such as the age of the concrete at the time of the material test (carried out shortly after the torsional tests), the cubic ($f_{cm,cube}$) and cylindrical ($f_{cm,cyl}$) compressive strengths, and the tensile strength ($f_{ctm}$), of all individual components—is provided in Table 1. The values listed are the average values from the conducted material tests (i.e., three cubic compression tests, six cylindrical compression tests, and three splitting tests for tensile strength). Tensile strength tests from previous test series indicated that the yield plateau for the 8, 10, and 12 mm diameter reinforcements was reached at stresses in the range of 640–680 N/mm$^2$. Although these material tests are from previous batches, experience has shown that they rarely deviate from this range of values.

**Table 1.** Material properties of the different components of the two test specimens.

| Concrete Mixture | Concrete Age (Days) | $f_{cm,cube}$ (N/mm$^2$) | $f_{cm,cyl}$ (N/mm$^2$) | $f_{ctm}$ (N/mm$^2$) | Associated Components |
|---|---|---|---|---|---|
| 1 | 41 | 61.2 (±1.0%) | 49.5 (±1.5%) | 3.32 (±8.1%) | TS$_{Ref}$: webs |
| 2 | 30 | 60.8 (±2.7%) | 50.2 (±4.7%) | 2.91 (±10.6%) | TS$_{Ref}$: bottom slab |
| 3 | 28 | 74.5 (±3.7%) | 64.3 (±5.3%) | 3.19 (±8.4%) | TS$_{Ref}$: deck slab |
| 4 | 44 | 62.5 (±1.8%) | 52.3 (±2.0%) | 3.23 (±9.6%) | TS$_{Joint}$: webs |
| 5 | 40 | 65.8 (±0.2%) | 55.1 (±2.7%) | 3.59 (±5.8%) | TS$_{Joint}$: bottom slab |
| 6 | 38 | 80.9 (±6.7%) | 64.7 (±3.2%) | 3.59 (±7.9%) | TS$_{Joint}$: deck slab |

### 2.2. Methodology: Test Setup, Load Application, and Measurement Setup

The test setup required to create the static system described in Section 2.1 is shown in Figure 5.

In a first step, the test specimen was vertically prestressed at the three locations of the diaphragms. This was carried out by using double-C steel beams on the top and bottom of the beam, through which threaded rods were passed, tensioned, and anchored. This step was necessary to avoid gaps between the steel beams and the specimen during torsional loading—applying in particular to the load application area, where a preload of up to 250 kN was generated by means of hydraulic presses. Next, the prepared test specimen was placed onto load cells (see Sections B–B and C–C), which, in turn, were placed on steel beams. As the restraint was to be created in the center of the girder (i.e., Section B–B), the load cells and steel beams were placed as far out as possible, in order to increase the lever arm and, thus, reduce the occurring forces. To generate the restraint, the test specimens were anchored against the strong floor of the test area (Section B–B), where the middle diaphragm (representing the pier segment of the new construction method) was situated. This was also carried out with threaded rods, which were passed through the double-C beams and anchored to the floor of the testing area. The rods were then prestressed to 350 kN by means of hydraulic presses. This not only created the restraint needed to take on the torsional load, but also simulated the vertical forces from the anchorage of the post-tensioning tendons that would arise in a real bridge project. The force was chosen so that, scaled to the size of the test body, it corresponded to the deviation forces of the post-tensioning tendons of a bridge project designed alternatively with the new method. To prevent tipping, threaded rods were also attached to the unloaded ends of the test specimens (Section C–C); however, these were only tensioned by hand to 60 kN, resulting in a small torsional moment transmitted to the unloaded area of the beam. As this torsional moment was of small magnitude and no torsional cracks occurred in this area, any influence on the following torsion test could be excluded. To apply the torsional

load (Figure 5a,b: Section A–A; Figure 6a), a press was first placed between the lower steel girder and the bottom of the testing area. This press was hydraulically coupled with another press, which was placed on the opposite side of the cross-section at the top of the steel girder, so that exactly the same pressure—and, thus, force—was applied at all times. The activation of the two presses created a torsional moment that twisted the girder around its axis. Large bending of the threaded rod was prevented by a calotte in the press assembly. As the torsional load increased, the lower press tended to slide laterally outwards. This displacement was made possible by introducing a sliding surface (PTFE) between the steel plates and the press, in order to avoid undesirable stresses. The displacement was kept low by using a wedge plate positioned between the press and the upper steel beam.

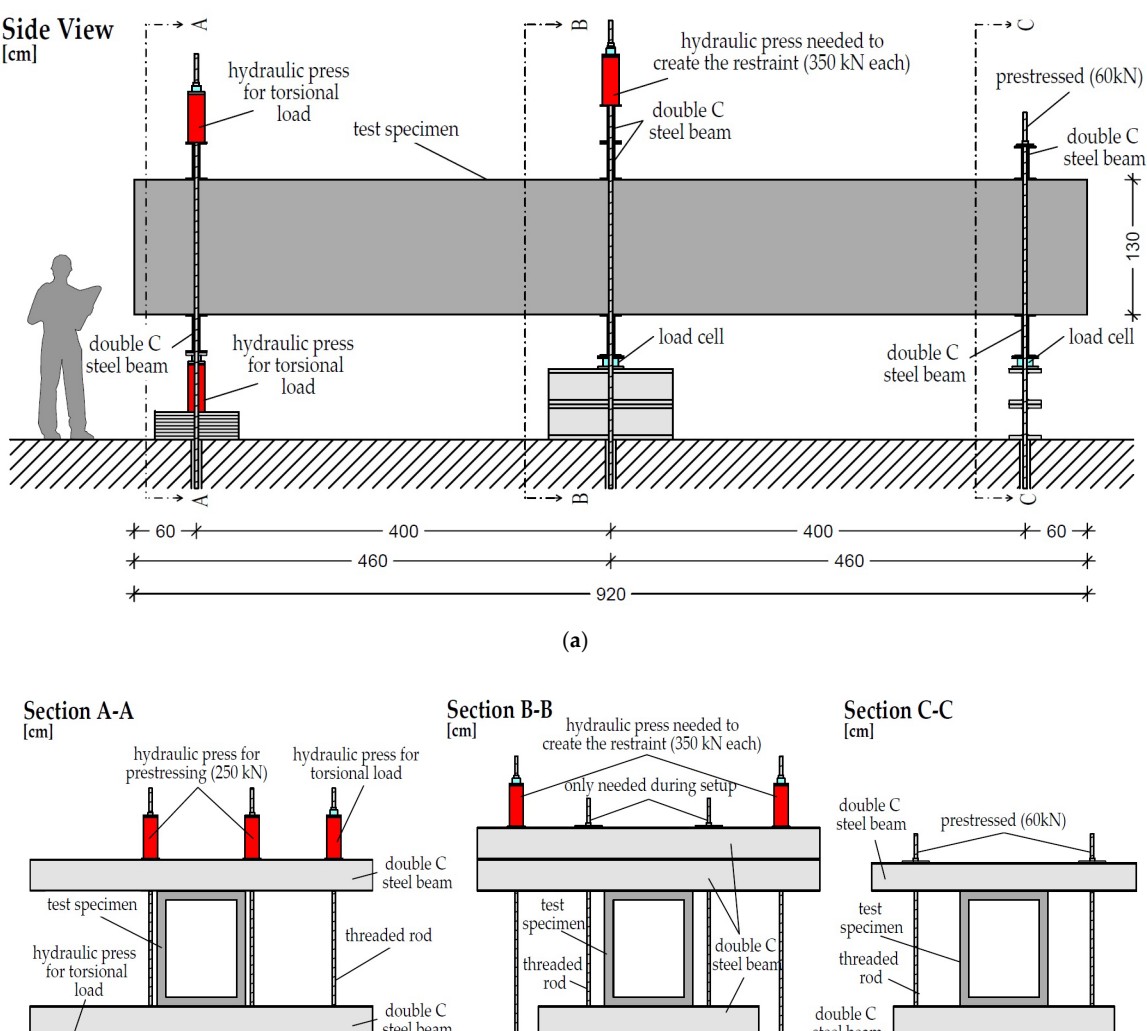

**Figure 5.** Illustration of the test setup: (**a**) view of the test body from the side onto the web. (**b**) Section A–A: torsional load application area; Section B–B: prestressing against the strong floor using hydraulic presses to achieve the restraint; Section C–C: additional restraint to avoid tipping of the specimen.

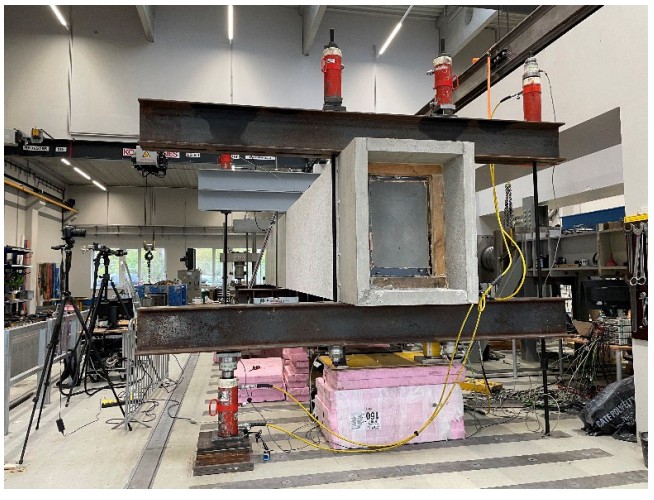

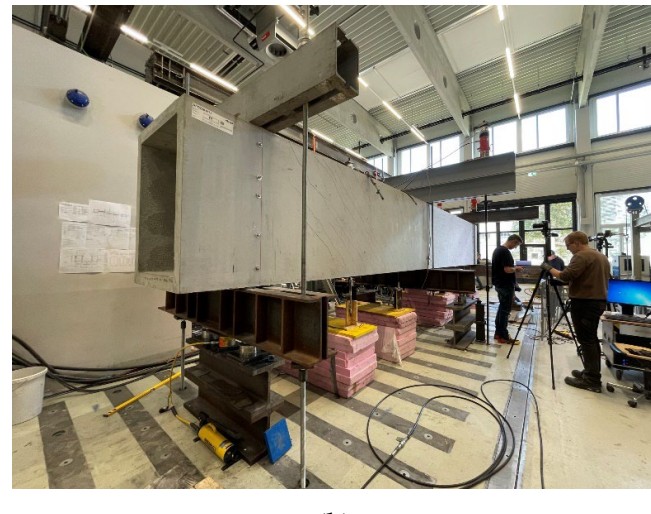

(**a**)

(**b**)

**Figure 6.** Test setup and measuring equipment: (**a**) View of the load application area (Section A–A). Hydraulic presses in red, including loading cells. (**b**) View of Sections B–B and C–C. Transducers placed on yellow wooden panels to measure the deflection of the webs.

The measuring equipment consisted of load cells (Lorenz K18 1000 kN and HBM RTN100T/C3 1000 kN (Lorenz Messtechnik GmbH, Alfdorf, Germany), maximum relative error of indication < ±1%, measurement frequency 5 Hz) (Figure 6a) used to document the reaction forces in Sections B–B and C–C, as well as all forces in the used hydraulic presses. To measure displacements, transducers (Solartron BS50 100 mm (Solartron Metrology, West Sussex, UK), maximum relative error of indication < ±1%, measurement frequency 5 Hz) were attached to the undersides of both webs (Figure 6b). An optical measuring system (Aramis 4M, resolution 2352/1728 pixels, GOM, measurement frequency 1/4 Hz) was used for crack documentation (Figure 6a). As the area of and around the vertical joint of $TS_{Joint}$ was of particular interest, this area was also recorded using the optical measuring system. The lateral displacement of the lower press was determined using transducers (Schreiber SM222.20.2SX79 20 mm (Schreiber Messtechnik GmbH, Oberhaching, Germany), maximum relative error of indication <±1%, measurement frequency 5 Hz).

For the test sequence itself, the force in the two presses of Section A–A was increased until torsional failure of the beam occurred. After documentation, the body was then rotated and reinstalled for the next test.

*2.3. Results*

As mentioned in the previous sections, a total of four torsion tests were carried out. The reference test specimen $TS_{Ref}$, was produced with one of the two outer diaphragms made from wood instead of mortar. As this allowed for too large cross-sectional distortions, a significantly earlier failure occurred, making the results irrelevant for the discussion. For this reason, the results of only one test are presented for the $TS_{Ref}$ specimen—namely, the one with the mortar diaphragm. With reference to $TS_{Joint}$, both tests provided relevant results and are documented below.

2.3.1. Course of Experiments and Maximal Torsional Moments

The maximum torsional moment achieved highlighted the differences between the two test specimen variants. Figure 7 shows the torsional moment curve as a function of the cross-sectional angle of twist measured at the load application point (Section A–A) for the three viable tests. The torsional moment was calculated as the measured force in the hydraulically coupled presses multiplied by their distance from one another. The displacement of the lower press (approx. 2–5 cm), as explained in Section 2.2, had to be added to the planned distance (300 cm) between the two presses.

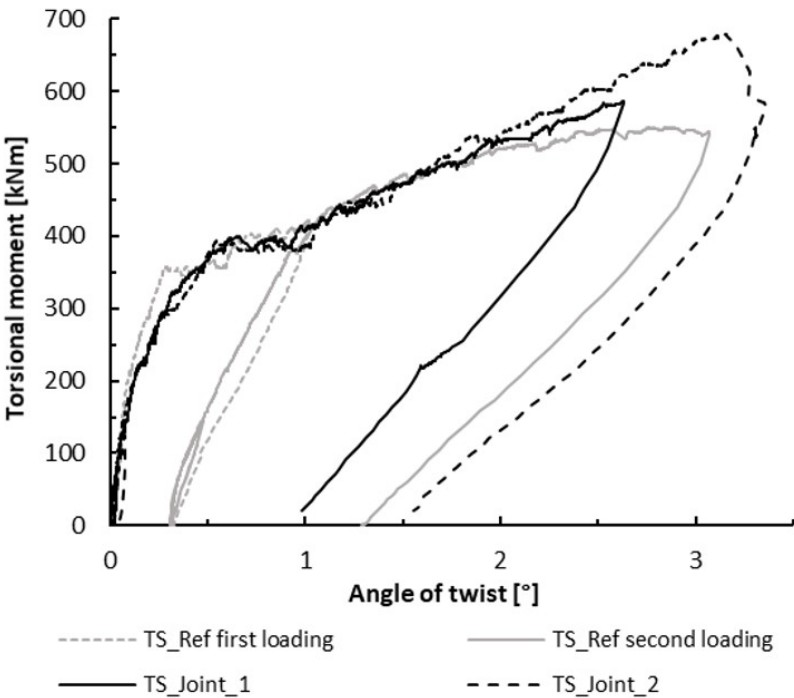

**Figure 7.** Torsional moment plotted against the angle of twist.

In the course of initially loading TS$_{Ref}$, a gap was observed between the test specimen and the steel girders at the load application point at a torsional moment of 420 kNm. This was due to the insufficient preload (150 kN/threaded rod). It was decided to unload the test specimen, increase the prestressing to 250 kN/threaded rod, and restart the test, explaining the division of the curves in Figure 7 into TS_Ref first loading and TS_Ref second loading. The initial crack within the specimen occurred at around 375 kNm. At an almost-constant load, a complete crack pattern formed at this point; thus, the transition to state II was complete. The load could then be further increased until a maximum value of 551 kNm. As a yield plateau was reached and no further load increase was possible yet the twist angle steadily increased, the test was stopped.

Within the TS_Joint_1 test, the initial crack and subsequent transition to state II occurred at around 380 kNm. In contrast to TS_Ref, no yield plateau occurred after further load increase. At around 580 kNm, the concrete began to spall due to compression in the area of the middle diaphragm, without any major load drop. A further increase in the load was considered possible; however, it was decided to stop the test at this point to prevent damage to the second test area. The test TS_Joint_2 performed almost identically up to the maximum load of TS_Joint_1, but without spalling. At a load of 678 kNm, sudden failure of the unreinforced vertical shear joint between the web and the diaphragm occurred.

### 2.3.2. Crack Patterns

The crack pattern results were derived from measurements taken on the specimens themselves, photos taken by the optical measuring system, and additional photos taken in the areas not covered by the optical measuring system. A compilation of the crack patterns is shown in Figure 8. As TS_Joint_1 was not tested until failure, yet the crack pattern (crack inclination) was very similar to that of TS_Joint_2, only the results of the latter are shown. The crack inclination angle α, as defined in Figure 8a, was determined by linearly connecting the start and end points of a crack and then determining their inclination with respect to the longitudinal axis. As most cracks had an approximately straight course, this procedure seemed to be rational. For cracks with more pronounced curvature, the crack was divided into subsections, and the mean angle of the individual partial lines was used. The crack with the largest width (i.e., the one ultimately resulting in failure) was marked

with a thicker line. In general, the mean crack inclination angles $\alpha_{avg}$ on the webs and bottom plate for all test specimens ranged from 47° to 51°, while those of the top plates were steeper, reaching values of 52° to 55°. The two almost-vertical cracks in Figure 8b were already formed when the specimen was lifted; therefore, they were not considered for the calculation of $\alpha_{avg}$. In addition to the $\alpha_{avg}$, the steepest angle $\alpha_{max}$ and flattest angle $\alpha_{min}$ are listed for each surface.

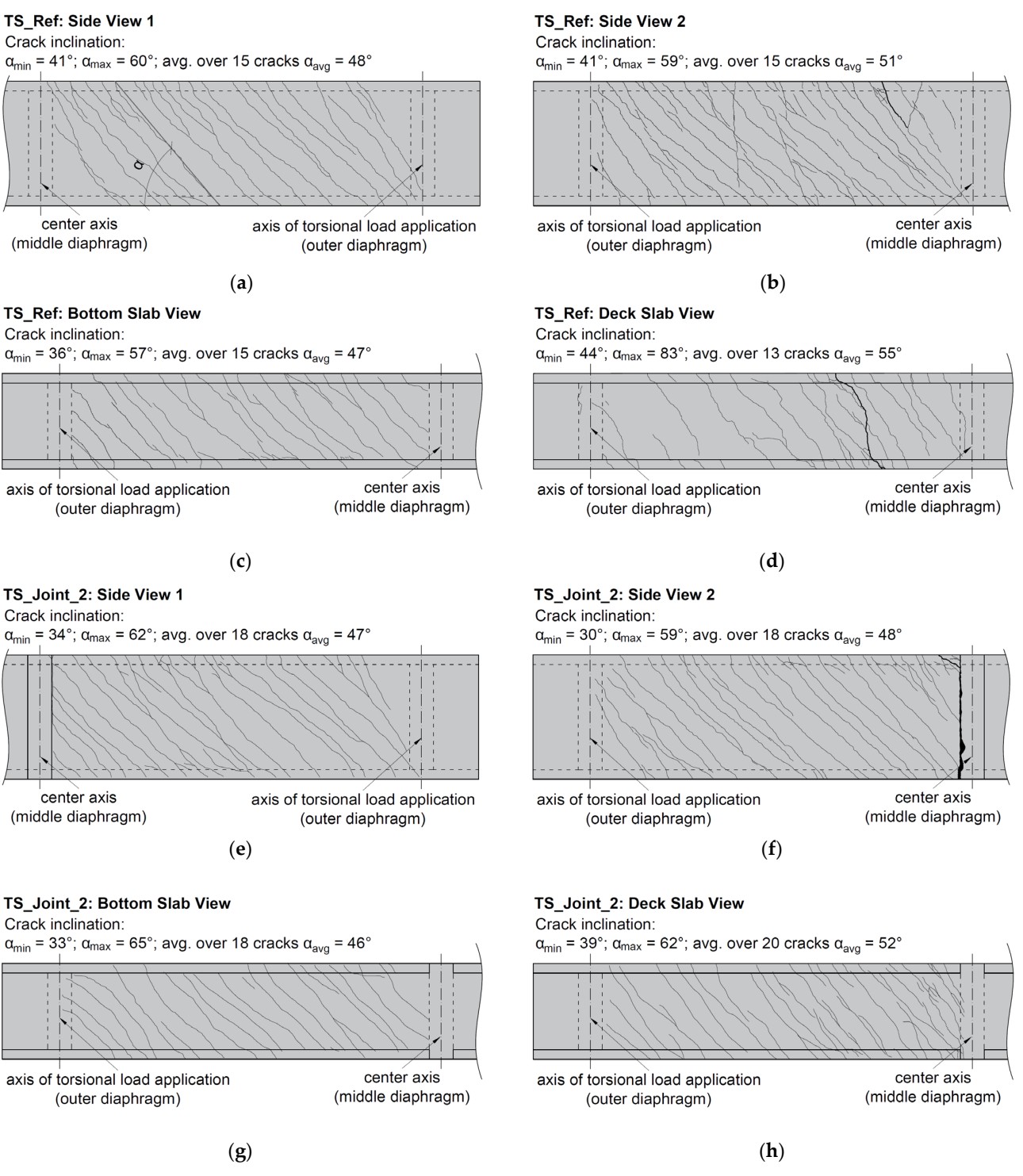

**Figure 8.** Crack patterns for TS_Ref and TS_Joint_2: (**a**) TS_Ref, side view 1; (**b**) TS_Ref, side view 2; (**c**) TS_Ref, bottom slab view; (**d**) TS_Ref, deck slab view; (**e**) TS_Joint_2, side view 1; (**f**) TS_Joint_2, side view 2; (**g**) TS_Joint_2, bottom slab view; (**h**) TS_Joint_2, deck slab view.

### 3. Comparison of the Experiments with Analytical Models and Discussion of the Results

*3.1. Calculation for Pure Torsion and Combination of Torsion and Bending*

After testing, we attempted to confirm the experimental results mathematically using the truss model found in common standards (e.g., Eurocode 2 [15]). Here, the stirrup reinforcement (1) and the longitudinal reinforcement (2) for pure torsion are determined separately:

$$\frac{A_{sw}}{s_w} = \frac{T}{2 \cdot A_k \cdot f_{y,s}} \cdot tan(\theta), \tag{1}$$

$$\frac{A_{sl}}{u_k} = \frac{T}{2 \cdot A_k \cdot f_{y,l}} \cdot cot(\theta), \tag{2}$$

where $T$ is the torsional moment; $A_{sw}$ is the diameter of a stirrup; $s_w$ is the distance between the stirrups in the longitudinal direction of the beam; $A_{sl}$ is the total area of longitudinal reinforcement installed in the beam; $u_k$ *is the circumference of the centerline of the equivalent box girder, which coincides with the real cross-section for the tests performed*; $A_k$ is the area inside the centerlines of the equivalent box girder; $f_{y,s}$ and $f_{y,l}$ are the yield stresses of the stirrup (*s*) and the longitudinal reinforcement (*l*), respectively; and $\theta$ is the inclination of the compression strut. Table 2 lists the parameters of the two test specimens for pure torsion. For TS$_{Joint}$, two calculation variants were analyzed: one considering only the continuous longitudinal reinforcement in the deck and bottom slabs within $A_{sl}$ (representing the cross-section at the restraint), and the second also considering the longitudinal reinforcement of the webs ending at the joint (representing the cross-section in the field).

**Table 2.** Parameters for pure torsion.

| Test Specimen | $A_{sw}$ (cm$^2$) | $s_w$ (cm) | $A_{sl}$ (cm$^2$) | $u_k$ (cm) | $A_k$ (cm$^2$) | $f_{y,s}$ (N/mm$^2$) | $f_{y,l}$ (N/mm$^2$) |
|---|---|---|---|---|---|---|---|
| TS$_{Ref}$ | 0.79 | 15.0 | 14.07 | 420.0 | 1080.0 | 680 | 652 |
| TS$_{Joint}$ w/o web-reinf. | 0.79 | 15.0 | 14.07 | 420.0 | 1080.0 | 680 | 652 |
| TS$_{Joint}$ w/web-reinf. | 0.79 | 15.0 | 23.12 | 420.0 | 1080.0 | 680 | 652 |

The fact that the ultimate load capacity has not yet been reached when the yield point of one of the two reinforcements is reached (cf. Section 1) was previously described by Lampert et al. in 1968 [16]. The force is redirected by a change in the inclination of the compression struts to the reinforcement that has not yet yielded. This is considered by equating the maximum torsional moment due to longitudinal reinforcement with that of the stirrup reinforcement, and then solving the equation according to the compression strut inclination (3):

$$\theta = tan^{-1} \left( \sqrt{\frac{\frac{A_{sw} \cdot f_{y,s}}{s_w}}{\frac{A_{sl} \cdot f_{y,l}}{u_k}}} \right). \tag{3}$$

If the compression strut inclination obtained from Equation (3) is substituted into Equation (1) or (2) and the equation is solved for $T$, a much more accurate value—when compared to the real ultimate torsion load—can be obtained, even though certain limits are set regarding the force redistribution [15,19].

An extension to the equations for the combined loading from torsion and bending was developed in 1970 [17,20]. In order to keep the calculation simple, the test specimens investigated by Lampert and Thürlimann [17,20] were converted into cross-sections that only had reinforcement in the corners. The resulting minor inaccuracy with respect to the lever arm for the bending moment can be considered acceptable. In order to compare the experiments determined here with those investigated by Lampert and Thürlimann [17,20], the same procedure was followed. The 28 Ø8 mm longitudinal bars of TS$_{Ref}$ were thus converted into corner bars with an equivalent area of 7 Ø8 mm each. In TS$_{Joint}$, a distinction

was again made between the field section and the restraint, resulting in corner bars with an equivalent area of 7 Ø8 mm + 2 Ø12 mm in the field section and 2.5 Ø8 mm + 2 Ø12 mm (=7 Ø8 mm) for the restraint. This area had to be reduced by the area required to take on the bending moment. For this purpose, the lever arm was selected as described in [16,17,20] and defined as the distance between the chord lines (i.e., at 1.20 m). The weakest side of a cross-section is decisive for failure [16,17,20]. In the case shown here, this concerns the deck slab, which is under tensile stress due to torsion and bending. In order to continue using Equations (1)–(3), the corner bars in the bottom slab—which are actually under compressive stress as a result of bending—had to be reduced by the same amount. Alternatively, Equations (1)–(3) could also be adapted to determine the torsional resistance of each individual slab, which would ultimately lead to the same result. The first approach was chosen, and the reduced area $A_{sl,red}$ was used to replace $A_{sl}$ in Equations (1)–(3). The occurring bending moment to be considered—resulting from the cantilever system of the test setup (the dead weight of the box girder, the diaphragm at the load introduction point, the steel girders, and the presses)—differed for the two specimens. While the moment at the crack leading to failure was chosen for $TS_{Ref}$, the maximum bending moment at the restraint was used for $TS_{Joint}$. The parameters for the calculation of the combined loading of torsion and bending resulting from the previously discussed assumptions are listed in Table 3. The values for $A_{sw}$, $s_w$, $u_k$, $A_k$, $f_{y,s}$, and $f_{y,l}$ can be taken directly from Table 2 as no alteration were made.

**Table 3.** Additional parameters needed for combined torsion and bending.

| Test Specimen | $M_{experiment}$ (kNm) | $A_{s,corner}$ (cm$^2$) | $A_{s,corner,red}$ (cm$^2$) | $A_{sl,red} = 4 \cdot A_{sl,corner,red}$ (cm$^2$) |
|---|---|---|---|---|
| $TS_{Ref}$ | 96 | 3.52 | 2.91 | 11.62 |
| $TS_{Joint}$ w/o web-reinf. | 161 | 3.52 | 2.49 | 9.96 |
| $TS_{Joint}$ w/web-reinf. | 161 | 5.78 | 4.75 | 19.01 |

For $TS_{Joint}$, the shear joint can be regarded as an upper boundary for the maximum torsional moment. The stress is determined by means of the first Bredt's formula added to the average shear stress resulting from the shear force (4):

$$\tau_{web} = \frac{V_{experiment}}{2 \cdot h_{web} \cdot b_{web}} + \frac{T_{experiment}}{2 \cdot A_k \cdot b_{web}} \; . \tag{4}$$

The calculated results for all variants under the load case torsion and bending are listed in Table 4 and are directly compared to the results obtained from the experimental tests. The shear stress $\tau_{web}$ was evaluated at the restraint for all variants.

**Table 4.** Comparison of the computational analysis with the experimental results for the load case torsion and bending.

| Test Specimen | $T_{max,experiment}$ (kNm) | $M_{experiment}$ (kNm) | $V_{max,experiment}$ (kN) | $\Theta$ (°) | $T_{max,calculated}$ (kNm) | $T_{max,experiment}/$ $T_{max,calculated}$ | $\tau_{web}$ (N/mm$^2$) |
|---|---|---|---|---|---|---|---|
| $TS_{Ref}$ | 551 | 96 | 61 | 54.6 | 548 | 1.01 | 2.80 |
| $TS_{Joint}$ w/o web-reinf. | 678 | 161 | 61 | 56.6 | 507 | 1.34 | 3.39 |
| $TS_{Joint}$ w/web-reinf. | 678 | 161 | 61 | 47.7 | 700 | 0.97 | 3.39 |

### 3.2. Discussion and Interpretation of the Results

For the analysis of the results, the test specimen $TS_{Ref}$ is discussed first. In addition to the results, Table 4 shows the ratio of the measured torsional moment in relation to the calculated results, with the deviation of only 1% demonstrating a particularly good agreement. This confirms the force redistribution from the longitudinal bars (which began to yield first) to the stirrups, accompanied by the inclination of the compression struts from

45° to the calculated 54.6°. It is important to note that the failure did not occur at the place with the highest bending moment—the restraint (M = 161 kN)—but rather at a distance of 1.20 m (M = 96 kN). The rationale for this behavior, as described below, is of utmost relevance for the further understanding of the load-bearing behavior of $TS_{Joint}$.

According to Lampert et al. [16,17,20], it is not sufficient for one side of a box girder to have reached its torsional load capacity to lead to a failure of the entire girder, as the load-bearing reserves of the remaining sides of the box girder are activated. This theorem was proven by Lampert and Thürlimann [16,17,20] by comparing girders with bending reinforcement on one side with ones with distributed reinforcement around the cross-section. Additional load reserves in the girder with bending reinforcement, with regard to torsion, were determined, allowing for an increase in load beyond the theoretically possible, according to the truss model. Coming back to $TS_{Ref}$, with equal longitudinal reinforcement around the cross-section, other load reserves must have been active in order for the girder not to fail at the restraint. A plausible explanation could be the vertical compressive stress in the webs, resulting from two sources: the downward tensioning of the middle diaphragm simulating the vertical forces of the prestressing, and the reaction forces resulting from the concrete compression struts at the restraint. As these compressive stresses decrease with increasing distance from the restraint, a failure in the field occurs in accordance with the truss model.

With regard to $TS_{Joint}$, the field section is considered first, i.e., the area where the effectiveness of the longitudinal reinforcement in the webs can be assumed ($TS_{Joint}$ w/web-reinf.). Table 4 shows that failure due to yielding of the reinforcement can be expected at 700 kNm. As the test specimen failed at 680 kNm due to shear failure in the joint, this value was not reached. The question still arises as to how such high values could be achieved at all if the longitudinal reinforcement ends directly at the restraint and, thus, is not supposed to be effective. The exact same explanation as for $TS_{Ref}$ must be considered. The vertical compressive stress must have been sufficient to strengthen the webs at the restraint to an extent of making it equal to the field section. In addition, the vertical compressive stress reduced the required anchorage length for the longitudinal reinforcement abutting the joint, making it fully effective after a short distance. However, the exact influence of the compressive stresses will have to be determined in further experimental investigations in order to make a definite statement. A possible test setup could be the placement of the joint directly in the field area. Similar tests, but with the joint reaching across the entire cross-section (i.e., including the bottom and top plates), are presented by Specker [25]. The maximum torsional load was limited by the shear stress in the joint, with an average calculated shear stress in the web of 3.39 MPa according to Equation (4) (without the consideration of a reduction due to the interlocked joint).

As indicated by the analysis results listed in Table 4, a redistribution of forces from the longitudinal reinforcement to the stirrups can be concluded for $TS_{Joint}$. In this context, it is important to mention that the redistribution was still achieved even though the additional longitudinal reinforcement (in the form of Ø12 mm bars) was not placed exactly in the corners. The plastic compression strut angle > 45° further indicates that both variants are above the load predicted by standards (e.g., Eurocode 2 [15]), where failure is assumed to occur when the yield stress of either the longitudinal or stirrup reinforcement is reached. In addition, it can be stated that the test specimen $TS_{Joint}$ was at least equivalent—and, in the present case, even superior—to the reference girder $TS_{Ref}$, due to the adapted reinforcement layout with additional corner bars in combination with the local vertical compressive stresses.

## 4. Conclusions

In this paper, investigations of the torsional load-bearing behavior of hollow box girders—which exhibit unique design properties due to the new LT-bridge construction method—are presented. A comparison was made between a reference girder, based on a conventional bridge design, and a girder designed according to the newly developed

bridge building method, incorporating vertical joints without longitudinal reinforcement in the webs directly at the connection to the pier segment. This comparison was made to distinguish the torsional load-bearing behavior of the novel bridge superstructure from that of conventional designs. Since the torsion tests found in the literature only consider hollow box girders without joints or girders in segmental construction with unreinforced joints, a new scientific investigation was necessary. The presented analyses are intended to lay the foundation for further application of the LT-bridge construction method. Within this paper, the experimental investigations were first presented before being verified using mathematical methods. The following conclusions were drawn from these investigations:

- The reference test specimen $TS_{Ref}$ failed at a torsional moment of 551 kNm and an associated bending moment of 96 kNm, with the decisive crack occurring at a distance of 1.20 m from the restraint. The computational analysis presented almost exact agreement with the experimental investigations. However, it was observed that the girder did not fail at the restraint, where the associated bending moment was significantly higher (i.e., 161 kNm). It can be assumed that the vertical compressive stresses in the web—resulting on the one hand from the simulated deflection force of the prestressing and, on the other hand, from the reaction forces resulting from the compression struts from the webs on the restraint—increase the torsional load-bearing capacity.
- The test specimen $TS_{Joint}$, which represents the bridge superstructure of the new bridge construction method, was able to withstand higher torsional loads (i.e., up to 681 kNm) than the reference girder. This was only due to the adapted reinforcement layout with additional bars in the corner areas of the deck and bottom slabs. The fact that no premature failure occurred, despite the unreinforced joints at the connection to the restraint, can again be attributed to the vertical compressive stresses in the joints and webs which, on the one hand, hinder crack opening due to torsion and, on the other hand, ensure a shorter anchorage length of the longitudinal reinforcement abutting the joint.
- Further investigations will focus on the influence of the vertical compressive stresses. Thus, for the next test series, it is planned to move the joint into the span section.
- Another important point to be analyzed in future experimental studies is the influence of post-tensioning on the torsional behavior of girders according to the LT-bridge construction method. Since this investigation is beyond the scope of this paper, the analysis of the effects of post-tensioning was excluded for the time being. However, experiments with longitudinal post-tensioning will be carried out in the near future.

**Author Contributions:** Development of new bridge construction method, J.K., F.U. and M.R.; torsion tests on hollow box girders, test conceptualization, M.R., F.U. and J.K.; methodology, M.R.; investigation, M.R.; analysis, M.R.; writing, M.R.; supervision, J.K. All authors have read and agreed to the published version of the manuscript.

**Funding:** This research was funded by FFG, grant number 880272. Open-access funding by TU Wien.

**Institutional Review Board Statement:** Not applicable.

**Informed Consent Statement:** Not applicable.

**Data Availability Statement:** The data that support the findings of this study are available from the corresponding author upon reasonable request.

**Acknowledgments:** The research project "Bridge construction with thin-walled segments out of pre-fabricated elements" was organized by the Austrian Society for Construction Technology (ÖBV) and financially supported by the Austrian Research Promotion Agency (FFG) through the following companies: ÖBB, ASFINAG, Porr, Strabag, Swietelsky, Habau, Implenia, Hochtief, Zeman, Östu-Stettin, Leyrer & Graf, Oberndorfer, ANP-Systems, VÖB, VÖZ, FCP, Baucon, Schimetta, Öhlinger & Partner, ZT Mayer. The authors express their sincere gratitude for the financial support. The authors are thankful for the open-access funding by TU Wien.

**Conflicts of Interest:** The authors declare no conflict of interest.

## Abbreviations

| | |
|---|---|
| $\alpha_{avg}$ | Average crack angle (°) |
| $\alpha_{max}$ | Steepest crack angle (°) |
| $\alpha_{min}$ | Flattest crack angle (°) |
| $A_k$ | Area inside the centerlines of the equivalent box girder (cm$^2$) |
| $A_{s,corner}$ | Area of the longitudinal reinforcement in the corner (cm$^2$) |
| $A_{s,corner,red}$ | Area of the longitudinal reinforcement available for torsion in the corner (cm$^2$) |
| $A_{sl}$ | Total area of longitudinal reinforcement installed in the beam (cm$^2$) |
| $A_{sl,red}$ | Total area of longitudinal reinforcement available for torsion (cm$^2$) |
| $A_{sw}$ | Diameter of the stirrup (cm$^2$) |
| $b_{web}$ | Width of the web (m) |
| $f_{yl}$ | Yield stress of the longitudinal reinforcement (N/mm$^2$) |
| $f_{ys}$ | Yield stress of the stirrup (N/mm$^2$) |
| $h_{web}$ | Height of the web (m) |
| $M_{experiment}$ | Bending moment occurring in the experimental investigations (kNm) |
| $s_w$ | Distance between the stirrups in longitudinal direction of the beam (cm) |
| $T_{max,calculated}$ | Maximum torsional moment according to the calculations (kNm) |
| $T_{max,experiment}$ | Maximum torsional moment assessed within the experimental investigations (kNm) |
| $\tau_{web}$ | Shear stress in the webs (N/mm$^2$) |
| $\theta$ | Inclination of the compression strut (°) |
| $u_k$ | Circumference of the centerline of the equivalent box girder (cm) |
| $V_{experiment}$ | Shear force occurring in the experimental investigations (kN) |
| $V_{max,experiment}$ | Maximum shear force occurring in the experimental investigations (kN) |

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
