# Peer review of "On the Torsional Behavior of the Longitudinal Bridge Girders Used in the LT-Bridge Construction Method"

_applsci, doi:10.3390/app13116657_

Round 1
Reviewer 1 Report
The article deals with the testing of bridge structure elements used in a new construction method. The authors consider an important problem, hence it seems be interesting and useful for many researchers. The article is extensive and its structure is correct. The abstract contains all the necessary information presented in an essential form. The introduction is references mostly recent works of the Authors of the manuscript as well as only a few older articles. That makes the originality and novelty of the research could be better exposed. In order to improve the manuscript the authors are requested to refer to the similar articles of the other researchers from the recent years. This is missing in the manuscript.
Further on, the authors presented the background of their considerations, assumptions for the research and all necessary information on the methods applied. Later in the manuscript, the authors presented the results of the research. The presentation of the results is clear. The authors correctly analyzed the results. The approach and the results presented in the study look credible. The work ends with accurate conclusions of significant value. The paper has been prepared very carefully, and certainly is worth to be published after the improvement mentioned above.
The language used in the work is correct. There are no problems with understanding the intentions of the authors.
Reviewer 2 Report
The reviewer appreciates the tedious experimental laboratory work done by the authors. The technical contents of this article are interesting. And, the corresponding findings are useful information for the bridge engineering community. Nevertheless, I do not recommend its publication in the “Applied Sciences, MDPI” unless the following comments are taken into account within the text:
1) The authors’ contributions and findings should better be emphasized in the sections of abstract, introduction and conclusions.
2) Introduction. I suggest to the authors to mainly focus on the literature review which treats the structural behaviors of concrete girder bridges. The part that treats the LT-bridge construction method could be reduced (in fact, the corresponding patent has been cited). The peculiarity of this article is related to the torsional tests by considering that only a few works treated this behavior for concrete girder bridges in the past. Most published works focused on the flexural behavior of concrete girder bridges. Please refer to this last point through the following reference:
- https://doi.org/10.12989/sem.2018.67.3.255
3) Long-age conditions were not taken into account since the torsional tests were performed during the early-age of the concrete box-girder bridge specimens. Please refer to this point through the following reference:
- https://doi.org/10.1061/(ASCE)BE.1943-5592.0001210
4) Section 2.1: The authors omitted the post-tensioning during torsional tests. The post-tensioning force plays an important role in the torsional behavior of concrete box-girder bridges. This omission in the tests should clearly be specified within the text with the consequent effects on the experimental results.
5) The authors used conventional sensors (force transducer, dial indicators etc.). Please specify the corresponding characteristics of the devices and of the related measurements, e.g., frequency (or the period) of the recording data etc. Moreover, range, sensitivity, resolution and accuracy of the conventional sensors should be underlined.
6) An “Appendix” section, containing names and elaboration of the symbols used, should be inserted at the end of the article.
7) I suggest to the authors to edit all the text of the article with the help of a native English speaker. Grammar, punctuation, spelling, verb usage, sentence structure, conciseness, readability and writing style could be improved.
English language requires an extensive editing.
Reviewer 3 Report
This manuscript focuses on the torsional behavior of longitudinal bridge girders using the LT-Bridge construction method. The study has significant reference value for the rapid construction of bridges. While the topic is generally interesting and suitable for Applied Sciences, there are several key areas that require thorough revision and improvements:
1. It is recommended to provide a more detailed introduction to the LT bridge construction method, emphasizing its advantages over traditional bridge construction methods. This will help readers understand the unique features and benefits of the LT-Bridge method.
2. Please include additional information about the experimental devices used, such as the specific size and type of testing equipment. This will enhance the reproducibility and accuracy of the experiments conducted.
3. It is crucial to conduct a more comprehensive analysis and discussion of the results obtained from the LT bridge construction method. Explore the significance of these results in the context of bridge engineering and highlight their potential applications in practice.
4. In the conclusion section, it is stated that the addition of steel bars to the bridge deck and bottom plate, along with the placement of joints near supports/constraints, can enhance the cross-sectional torsional load capacity of the new construction method compared to conventional bridge superstructures. To support this claim, provide specific evidence or data that demonstrate the improved torsional load capacity achieved with the LT-Bridge method.
5. Consider including a cost comparison and evaluation of the economic benefits between the new LT-Bridge construction method and traditional methods mentioned in the paper. This will provide readers with insights into the potential cost savings or efficiency improvements associated with adopting the LT-Bridge method.
6. Address whether the experimental results might be affected by the omission of post tensioning and discuss any potential discrepancies between the experimental setup and real-world conditions. Clarify the potential impact of this omission on the interpretation of the results.
7. Discuss the applicability of the LT-Bridge method to concrete beams with other cross-sectional forms, such as complex box sections. Assess the adaptability and potential challenges associated with implementing the LT-Bridge method in various bridge designs.
Minor editing of English language required
Round 2
Reviewer 2 Report
The required revisions were carried out and the manuscript can be accepted for publication.
English language requires a moderate editing.
Reviewer 3 Report
The reviewer recommends the manuscript be accepted for publication due to the satisfactory responses to the previous comments.
Minor editing of English language required